# Reliability of Maximal Strength and Peak Rate of Force Development in a Portable Nordic Hamstrings Exercise Device

**DOI:** 10.3390/s23125452

**Published:** 2023-06-09

**Authors:** Júlio A. Costa, Konstantinos Spyrou, António Sancho, Joana F. Reis, João Brito

**Affiliations:** 1Portugal Football School, Portuguese Football Federation (FPF), 1495-433 Cruz Quebrada, Portugal; 2UCAM Research Center for High Performance Sport, UCAM Universidad Católica de Murcia, 30107 Murcia, Spain; 3Facultad de Deporte, UCAM Universidad Católica de Murcia, 30107 Murcia, Spain; 4Faculdade de Motricidade Humana, Universidade de Lisboa, 1495-751 Lisboa, Portugal; 5Interdisciplinary Center for the Study of Human Performance (CIPER), Faculdade de Motricidade Humana, Universidade de Lisboa, 1495-751 Lisboa, Portugal

**Keywords:** muscle strength, rehabilitation, groin, repeatability, reproducibility

## Abstract

The Nordic hamstring exercise (NHE) is a very popular exercise used to improve eccentric strength and prevent injuries. The aim of this investigation was to assess the reliability of a portable dynamometer that measures maximal strength (MS) and rate of force development (RFD) during the NHE. Seventeen physically active participants (34.8 ± 4.1 years; *n* = 2 women and *n* = 15 men) participated. Measurements occurred on two different days separated by 48–72 h. Test–retest reliability was calculated for bilateral MS and RFD. No significant test–retest differences were observed in NHE (test–retest [95% CI, confidence interval]) for MS [−19.2 N (−67.8; 29.4); *p* = 0.42] and RFD [−70.4 N·s^−1^ (−178.4; 37.8); *p* = 0.19]. MS showed high reliability (intraclass correlation coefficient [ICC] [95% CI], =0.93 [0.80–0.97] and large within-subject correlation between test and retest [r = 0.88 (0.68; 0.95)]. RFD displayed good reliability [ICC = 0.76 (0.35; 0.91)] and moderate within-subject correlation between test and retest [r = 0.63 (0.22; 0.85)]. Bilateral MS and RFD displayed a coefficient of variation of 3.4% and 4.6%, respectively, between tests. The standard error of measurement and the minimal detectable change for MS was 44.6 arbitrary units (a.u.) and 123.6 a.u., and 104.6 a.u. and 290.0 a.u. for peak RFD. This study shows that MS and RFD can be measured for NHE using a portable dynamometer. However, not all exercises are suitable to apply to determine RFD, so caution must be taken when analyzing RFD during NHE.

## 1. Introduction

Research into hamstring injuries has dramatically increased in the last two decades, because hamstring injuries are one of the most common injuries in high-speed running sports [1,2]. Specifically, one recent study conducted by Ekstrand et al. [1] reported that all hamstring injuries diagnosed in soccer in the 21-year study period have increased from 12% to 24%. Furthermore, the proportion of injury absence days caused by hamstring injuries increased from 10% to 20% [1]. Hamstring injuries are more likely to occur during running and sprinting, because the hamstring muscles experience the greatest amount of eccentric force during the late swing phase in the gait cycle [3], as the hip and knee muscles during late swing phase demonstrated the most dramatic increase in biomechanical load (i.e., torques, net powers, and work done) when running speed progressed [4,5,6]. Furthermore, hamstring injury depends on many factors [7]. Specifically, hamstring eccentric and concentric strength, lumbopelvic and knee stability, lower-limb stiffness, and insufficient sprint exposure may increase the likelihood of a hamstring injury occurring [7]. Given that, eccentric knee flexor muscle strength is one of the fundamental metrics to prevent injuries [8,9,10] and consequently increase an athlete’s specific performance (e.g., acceleration and high-speed running) [8,11,12].

The Nordic hamstring exercise (NHE) is one of the most common eccentric exercises used in sports and is currently used as a major injury prevention strategy [12]. The exercise instruction is the following: the athlete is asked to perform eccentric knee flexor maximal strength (MS) in a high kneeling position with the ankles fixed either by a partner or by a stationary object. From this position, the athlete inclines the torso, maintaining neutral hip alignment, for as far as possible and then uses the arms to contact the ground in front when the hamstrings can no longer control the downward movement. The NHE intervention seems to increase the fascicle strength by increasing the number of sarcomeres in series within the muscle fibers, and there are potential changes in the distribution through electromyography of the three biarticular of the hamstring muscles (i.e., biceps femoris long head, semitendinosus, and the semimembranosus) [13,14,15].

Regarding previous research, Lodge et al. [16] found high test–retest reliability, ICC 0.91 (CI, 0.76–0.96) and 0.91 (CI, 0.78–0.96) for left and right eccentric knee flexor muscle strength peak forces, respectively, using an eccentric hamstring strength measurement device similar to the portable dynamometer used in the current study compared to an isokinetic dynamometer. Furthermore, similar results were found in inter-rater reliability and correlations between isometric and eccentric knee extension and flexion strength using a hand-held dynamometry and isokinetic test for knee flexion extension. Consequently, it is vital to highlight that eccentric knee flexor muscle strength devices have already been validated, and the aim of this study was to evaluate the test–retest reliability of a portable dynamometer. To the authors’ knowledge, this is the first study that has measured reliability (i.e., test–retest) of the rate of force development (RFD) during knee flexor strength testing.

To date, the gold standard measure for the evaluation of eccentric knee flexor strength is isokinetic dynamometry [17]. However, isokinetic dynamometers are characterized by a lack of portability, high cost and time consumption, and their daily use might be practically difficult. Considering that a great number of devices that use load cell dynamometers have become popular field-based methods to monitor individual eccentric knee flexor strength during a NHE [9,17,18,19], therefore, the aim of this study was to evaluate the test–retest reliability of eccentric knee flexor MS and peak RFD during NHE using a portable dynamometer. The leading hypothesis of the current study was that the current portable dynamometer provides reliable data of eccentric knee flexor MS and RFD, and the study was designed to answer the main research question declared above.

## 2. Materials and Methods

### 2.1. Participants

Seventeen healthy and physically active adult subjects (*n* = 15 men and *n* = 2 women) who engaged in more than 3 h of physical exercise per week, who were injury free in the lower limbs, and had no pain or illness in the past 3 weeks before starting the study volunteered to take part in this study. Table 1 reports the characteristics of the participants. The experimental design and potential risks of the study were explained to the participants and written informed consent was provided. The study was approved by the Ethics Committee of the Portugal Football School, Portuguese Football Federation (CEPFS 12.2021).

### 2.2. Procedures

The same researcher recorded the test–retest NHE performance data at three distinct sessions on different days. Firstly, the participants performed a familiarization session that included the same warm-up, order, and exercises as the evaluation sessions. Approximately 7 days later, the participants performed the first test session, and the retest session was conducted within 48–72 h from the end of the first test session. All sessions were conducted at the same time of the day (i.e., during morning or afternoon). Participants were asked to not perform any vigorous lower-limb exercises in the 24 h before each testing session [20].

In the first session, the participants performed a warm-up that consisted of 7 min on a bicycle ergometer at a pedaling cadence of 75–80 rpm, 2 sets of 12 reps of half-squats, standing toe raises, and hip bridge [21]. According to recent literature [21], participants were positioned in a kneeling position over the padded board, ankles held under lockable braces (fixed atop the uniaxial load cells), with the lateral malleolus aligned with the edge of the board and arms across the chest, using the portable dynamometer (Figure 1).

The dynamometer Smart Nordic Trainer (Neuroexcellence^®^; S-2A INOX, Porto, Portugal) has two load cells (one on each hook to measure the force applied by each leg). Each cell has a maximum capacity of 4903 N (~500 kg). When starting the movement, the reading of the cell is correct, but in the middle of the exercise, the hook has a rotational movement of about 5 to 8 degrees, which is intentional, which is the adaptation of the hook to the athlete’s exercise, which can vary the angle from athlete to athlete. The manufacturers considered this read error to be negligible. A load of 100 kg with a hook rotated by 8 degrees corresponds to an error of ±1 kg. The cell reading is 100 g. Model SENSOCAR^®^ S-2A INOX has a repeatability error <0.02% F.E, sensitivity 2.0 mV/V ± 0.1%, zero offset < 1% F.E, combined maximum error < 0.02% F.E., Fluence 30 min (creep) < 0.02%. The metrics were calculated according to the manufacturer as follows:

N: number of recorded samples; F: Force list; t: Timestamp list

Maximal Force: MaxValue = max(F)
$$\mathrm{Maximal}\;\mathrm{Force}:\;\mathrm{MaxValue}\;=\;\max(F)\;\mathrm{Peak}\;\mathrm{RFD}=max\left(\left\{f\left(x\right):x\in \left[N..1\right]\right\}\right)\;f\left(x\right)=\frac{Fx-Fx-n}{tx-tx-n}$$
where n is the closest index and t_x_ − t_x−n_ is equal to the Time Interval RFD. Note: the default value of the Time Interval RFD is 0.05 s.

During the familiarization, and during test and retest sessions, the participants performed 3 maximal trials of eccentric NHE repetitions, where participants leaned forward in a slow, controlled manner for as long as possible, during the eccentric phase. The movement onset was determined by counting down from three to one (information given by the software), and then the participant started performing the NHE. Then, the participants passively returned to the starting position, in order to repeat the following repetitions. According to recent literature [21], the maximal NHE trials were separated by a standardized 1-min rest period to allow for recovery and to avoid fatigue. If participants increased their performance in all three trials, one or two additional NHE repetitions were performed [21]. Authorized feedback from the investigator was used to motivate the subjects. Trials were only regarded as successful if the participants held trunk and hips in a neutral position during the NHE repetitions. Participants controlled the movement until they lost control and stopped dealing with it. No additional loading was used. According to the ANHEQ criteria [22], the total score for NHE quality was 8 points, which is considered “good”. Specifically, ANHEQ criteria are the following: (1) Rigid fixation: 2 points; (2) Knee position: 1 point; (3) Kneeling height: 1 point; (4) Separate familiarization: 1 point; (5) Diagnostic tools: 1 point; (6) Feedback of target movement speed: 0 points—we only provided feedback to the participant to perform as slowly as possible; (7) Consequences of impaired technique: 1 point; (8) Presentation of NHE performance variables: 1 point. Bilateral MS and RFD were considered for analysis. All data were recorded with corporative data acquisition software (NexSo v1.0.0., Porto, Portugal). Data were collected through the Bluetooth BLE communication protocol at 180 Hz. The tests were performed in a gym facility.

### 2.3. Statistical Analyses

Sample distribution was tested using the Shapiro–Wilk test for MS and peak RFD variables. Variables are presented as mean with the 95% confidence interval (CI).

Linear mixed model analysis was performed to examine differences in the MS and peak RFD during test–retest.

To estimate the test–retest reliability of the NHE, intraclass correlation coefficients (ICC) [23] and the two-way random effects model of the measurements with 95% CI was used. The ICC were classified in the following manner: >0.90, high reliability; 0.80–0.89, good reliability; between 0.70 and 0.79, fair reliability; and values <0.69, poor reliability [24]. Further, within-subject variation was determined using typical error expressed as a coefficient of variation (CV) [25].

The standard error of measurement [25] and the minimal detectable change (MDC) were calculated to analyze the variability of the participants’ performances. For this analysis, the following formulas were used to calculate the SEM and MDC [25].
SEM = SD × √(1 − ICC)
MDC = SEM × √2 × 1.96

We tested the within-subject correlations (r, 95% CI) [26] between test and retest for MS and peak RFD variables. We qualitatively interpreted the magnitudes of correlation using the following criteria: trivial (r ≤ 0.1), small (r = 0.1–0.3), moderate (r = 0.3–0.5), large (r = 0.5–0.7), very large (r = 0.7–0.9), and almost perfect (r ≥ 0.9) [27].

Most of the statistical analyses were conducted using SPSS software (version 27.0.1, SPSS Inc., Chicago, IL, USA), except for within-subject correlation for which a rmcorr package in R statistical software (version 3.4.1, R Foundation for Statistical Computing, Vienna, Austria) was used.

## 3. Results

Values of bilateral absolute MS and RFD variables during the NHE in healthy and physical activity adults are presented in Figure 2.

During the familiarization session, MS was 669.4 N (581.1; 734.3) and peak RFD was 543.8 N·s^−1^ (428.7; 654.9). No significant test–retest differences were observed in NHE performance for MS and RFD variables (Table 2).

MS showed high reliability (ICC = 0.93 [0.80–0.97]) and large within-subject correlation between test and retest [r = 0.88 (0.68; 0.95)] (Figure 3). Peak RFD demonstrated good reliability [ICC = 0.76 (0.35; 0.91)] and moderate within-subject correlation between test and retest [r = 0.63 (0.22; 0.85)]. The MS and peak RFD presented CV values of 3.4% and 4.6%, respectively, between test and retest. The standard error of measurement and the MDC for MS was 44.6 arbitrary units (a.u.) and 123.6 a.u., and for peak RFD was 104.6 a.u. and 290.0 a.u.

Figure 4 and Figure 5 presents the force and time profile of the test–retest of a representative participant and the RFD in one repetition. Figure 6 depicts the force time profile of the first session of four randomly chosen participants in order to represent their different completions of the exercise.

## 4. Discussion

The aim of this study was to evaluate the test–retest reliability of the bilateral eccentric knee muscle flexor MS and RFD during NHE. The main findings were the following: (1) no significant test–retest differences were observed in NHE for MS and RFD; (2) MS showed *high* reliability and *large* within-subject correlation between test and retest; (3) RFD displayed *good* reliability and *moderate* within-subject correlation between test and retest; and (4) MS and RFD presented CV values of 3.4% and 4.6%, respectively, between test and retest. This study shows that MS and RFD can be measured for NHE using a portable dynamometer.

Regarding peak absolute strength, NHE showed *high* reliability (ICC = 0.93 and CV = 3.4%) and a *large* within-subject correlation between test and retest (r = 0.88) (Figure 3). The current results for test–retest reliability of the HSE device is in line with previous studies [9,17,19,28,29]. For example, Lodge et al. [17] found *high* test–retest reliability, ICC 0.91 (CI, 0.76–0.96), and 0.91 (CI, 0.78–0.96) for left and right eccentric knee flexor muscle strength peak forces, respectively, using an eccentric hamstring strength measurement device similar to the portable dynamometer used in the current study compared to an isokinetic dynamometer. Moreover, similar results showed an inter-rater reliability and correlations between the isometric and eccentric knee extension and flexion strengths using hand-held dynamometry and an isokinetic test for knee flexion extension of athletic participants. Therefore, it is vital to highlight that eccentric knee flexor muscle strength devices are already validated and the aim of this study was to evaluate the reliability of test–retest of a portable dynamometer. In a practical application, the current portable device can be used to evaluate and train eccentric flexor muscle strength on a daily basis.

Considering RFD showed *good* reliability (ICC = 0.76; 4.6%) and *moderate* within-subject correlation between test and retest (r = 0.63) (Figure 3). To the authors’ knowledge, this is the first study measuring the RFD reliability (i.e., test–retest) during a knee flexor strength test. Compared with the assessment of RFD for the hip muscles (i.e., hip adductor, flexor, and external rotator), RFD for the hamstrings can be measured with confidence (i.e., ICC  >  0.70 and standard error < 10%) [30]. Considering the *moderate* within-subject correlation between test and retest, it is important to highlight that RFD assessments might be challenging and need more time for familiarization with the test [31]; also, it is important to highlight that the NHE is not performed in a maximal isometric contraction, as it is in traditional RFD evaluations. Therefore, due to the controlled and slow movement in the NHE, the highest RFD may not occur at the beginning of the exercise. Therefore, RFD is acceptable to evaluate by the hamstring strength portable device, but it should be conducted with caution and familiarization. In an applied setting, the RFD value can be recorded from the portable dynamometer.

This study is limited by the NHE itself, as factors such as lack of control on the velocity of the movement, the intervention from other muscle groups, such as the lumbo-pelvic zone, and the determination of the “optimal” angle peak torque of the knee flexor muscle group, which would be useful when targeting strength improvements at a specific joint angle (that could be measured by the gold standard measurement such as an isokinetic dynamometer). Furthermore, regarding Assessing the NHE quality [21] scale knee position is a key component of NHE execution as, on a rigid surface, the pressure on the knees may cause an uncomfortable feeling and pain. Additionally, the RFD metric, even with *good* reliability [ICC = 0.76 (0.35; 0.91)] and *moderate* within-subject correlation between test and retest [r = 0.63 (0.22; 0.85)], should be used cautiously as it is a controlled and slow movement. The current study was designed to examine the test–retest reliability, considering essential to use on a daily basis, avoiding misrepresentation of changes in strength and minimizing the error of measurement. Further research about the validation of the current portable dynamometer when compared to gold standard measurements, such as isokinetic dynamometers or other similar portable devices that have already been validated, is warranted. Lastly, more investigation is warranted regarding NHE variations with rapid muscle activation, i.e., reactively bouncing and decelerating exercises which elicit much higher peak moments than the standard NHE [6].

## 5. Conclusions

In conclusion, the current device presented no significant test–retest differences during the NHE for MS and RFD. Furthermore, MS and RFD variables showed *good–high* reliability and *moderate–large* within-subject correlation between test and retest, respectively. Lastly, MS and RFD showed CV values of 3.4% and 4.6%, respectively, between test and retest. This study shows that MS can be measured during the NHE using a portable dynamometer, but this should be performed with caution and with previous familiarization due to the slow and controlled movement of the NHE that may not favor the attainment of the highest RFD at the beginning of exercise.

## Figures and Tables

**Figure 1 sensors-23-05452-f001:**
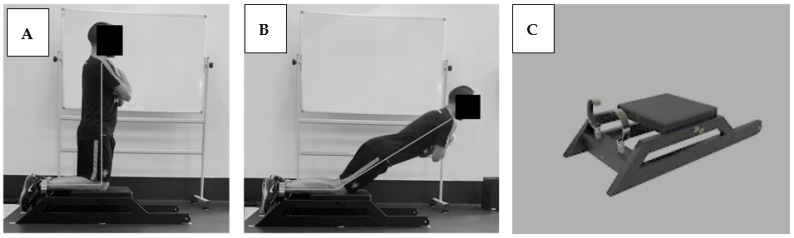
Testing set-up. (**A**) starting positioning; (**B**) participant leaned forward slowly and as controlled as possible (eccentric phase only); (**C**) portable dynamometer used in this study.

**Figure 2 sensors-23-05452-f002:**
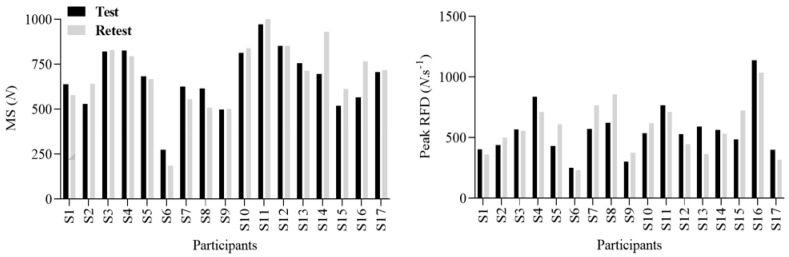
Descriptive test–retest individual data for MS and peak RFD during NHE in healthy and physically active adults (*n* = 17).

**Figure 3 sensors-23-05452-f003:**
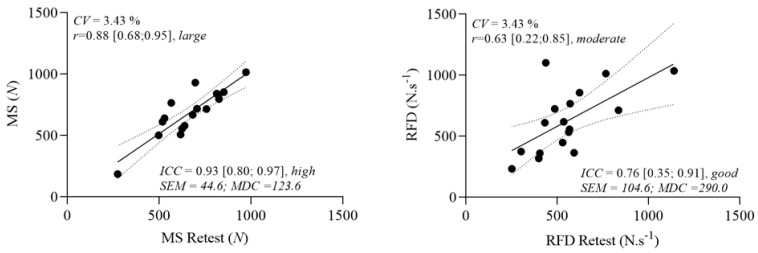
Test–retest reliability and a within-subject correlation was calculated for MS and peak RFD during NHE in healthy and physically active adults (*n* = 17). ICC, intraclass correlation coefficient [95%CI]; CV, coefficient of variation; SEM, standard error of measurement; MDC, minimal detectable change; N, Newton.

**Figure 4 sensors-23-05452-f004:**
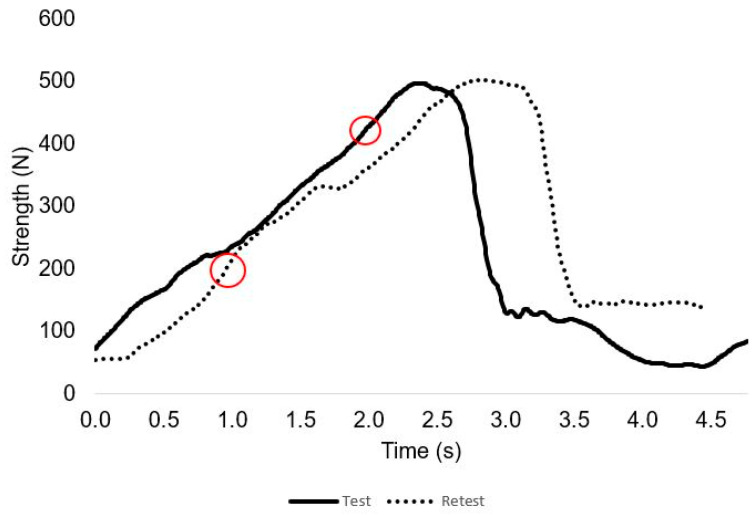
Test–retest individual data between strength (N) and time (ms) of a representative subject. Red circles symbolize the moment where the RFD was higher during NHE.

**Figure 5 sensors-23-05452-f005:**
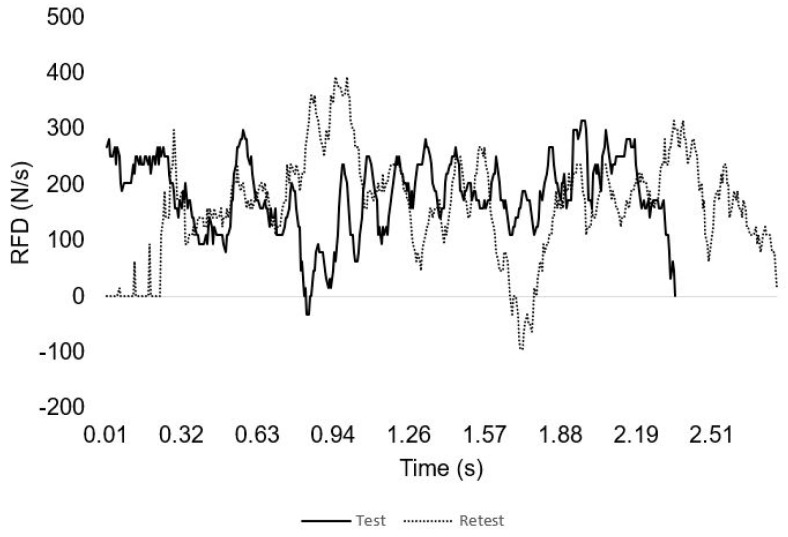
Test–retest individual data between RFD (N/s) and time (ms) of a representative subject.

**Figure 6 sensors-23-05452-f006:**
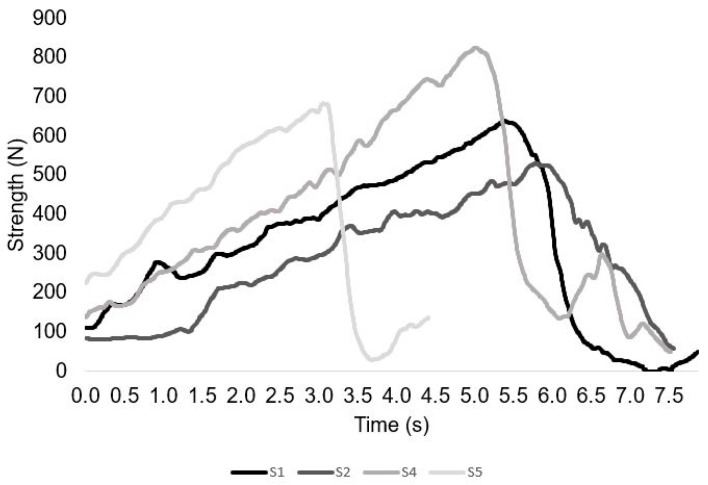
Example for different individual subjects (S1, S2, S3, S4, and S5) performing a NHE.

**Table 1 sensors-23-05452-t001:** Participants’ characteristics (*n* = 17).

	Total (*n* = 17)
Age (years)	34.8 ± 4.1
Body mass (kg)	78.5 ± 16.2
Height (m)	1.8 ± 0.1
BMI (kg/m^2^)	24.1 ± 3.6

Values are expressed in mean ± standard deviation. BMI, body max index.

**Table 2 sensors-23-05452-t002:** Descriptive and test–retest differences data for bilateral MS, relative MS, and peak RFD, during NHE in healthy and physically active adults (*n* = 17).

	Test	Retest	Δ (Test–Retest)	*p*
Bilateral MS (N)	669.8 (583.8; 755.9)	689 (588.3; 789.7)	−19.2 (−67.8; 29.4)	*p* = 0.42
Relative MS (N/Kg)	8.6 (7.7; 8.9)	8.7 (7.8; 9.6)	−0.1 (−0.8; 0.5)	*p* = 0.63
Peak RFD (N·s^−1^)	554.9 (446.6; 663.1)	625.3 (488.6; 761.9)	−70.4 (−178.4; 37.8)	*p* = 0.19

Values are expressed in mean (95% CI).

## Data Availability

Data may be available from the Data Protection Office, Portuguese Football Federation (Data Access contact via e-mail: dpo@fpf.pt) for researchers who meet the criteria to access confidential data.

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
