# Peer review of "Reliability of Maximal Strength and Peak Rate of Force Development in a Portable Nordic Hamstrings Exercise Device"

_sensors, 2023, doi:10.3390/s23125452_

Round 1
Reviewer 1 Report
Dear authors,
first of all I would like to thank you for your work and your prepared manuscript. However, in its current form it needs major revision to provide novel insights in the field of eccentric hamstring tests.
I will directly start with my specific comments because in my opinion addtional introductory words do not help to improve the quality.
GENERAL:
- I have great concerns about the use of RFD during dynamic movements, especially when the task should be performed in a slow controlled fashion. Right now, the use and interpretation of this parameter does not make sense. I am open to discuss this issue in more detail. However, please present some representative graphs of different participants first so that the reader can get an idea when the RDF occurred.
- Usually, RFD is derived from isometric tests and analysed within e.g. 100 ms. Please expand, why you chose a different methodology (50 ms time interval).
- Your work would dramatically benefit from including reliability and validity data from the force cells themselves (i.e. checking the calibration), especially with regard to potential effects of ankle hook tilt (or are your hooks rigid and therefore always perpendicular to the floor?; The Nordbord has this potential measuring inaccury, but nobody knows it ... even if a 10° tilt results in just a small deviation of 1.5%)
- Please consider to report the data of the familiarization session as well to emphasize a potential beneficial effect (i.e. significant differences) like it was demonstrated in isokinetic test scenarios (Variability of isokinetic measures; Factors influencing the reproducibility of isokinetic knee flexion and extension test findings).
- Weir (2005) recommended a three-layered approach for reliability studies than yours (Quantifying test-retest reliability using the intraclass correlation coefficient and the SEM). Please think about removing redundant parameters like CV and MDC as they just report the variability of the participants' performances instead of the device.
ABSTRACT:
l27: between tests
l28: arbitrary units (a.u.) -> please choose this order
l29: showed -> past tense, however this verb is used way toooo much throughout the manuscript. Please reduce its use!
INTRODUCTION:
l34: dramatically increased
l37: delete "last"
l40: delete "potentially"
l42: consider to add further research revealing why the late swing phase is injury-relevant, especially at highest running speeds (Effect of running speed on lower limb joint kinetics; The effect of speed and influence of individual muscles on hamstring mechanics during the swing phase of sprinting; Swing phase mechanics of maximal velocity sprints - Does isokinetic lower-limb muscle strength matter) -> include in the dicussion why NHE performance is not directly transferable to sprint mechanics
l43: delete "lower" -> lumbopelvic implies that it is the lower part of the back
l46: citations [5-7] are related to "prevent injuries" -> please modify the position of the citations
l46: specify performance (e.g. acceleration and high-speed running) and add related references (Acute adaptations and subsequent preservation of strength and speed measures following a Nordic hamstring curl intervention: a randomised controlled trial; Nordic Hamstring Exercise training induces improved lower-limb swing phase mechanics and sustained strength preservation in sprinters; Effects of the Nordic Hamstring exercise on sprint capacity in male football players: a randomized controlled trial)
l49: change to "most commonly used"
l56: please specify to which "distribution" you are referring -> EMG?
l58: Two studies are not "several" ... some -> however, I ask myself if this sentence is needed for the logical flow of your manuscript. Consider to remove it.
l62: remove "test"
l63: change to "their daily use might be difficult"
l67: you must be much clearer here! "Whilst peak force/moment is the most common parameter during NHE tests, the RFD has not been investigated yet ..." -> to be honest, I am not convinced that the RFD makes sense when you instruct a slow controlled NHE execution!
l68: please state that your hypothesis was that the measure are reliable
METHODS:
l74-75: no citations needed here
l95: be more clearer here: 1. familiarisation; 2. first test session; 3. second test session (retest)
l97: test session instead of "testing" -> here and throughout the entire manuscript
l98: change the order: "vigorous lower-limb exercises"
l111: change the order: "recorded samples"
Fig1: are the ankle hooks rigid or do they allow for a certain tilt with respect to the sagittal plane?
l134: "and" instead of "i.e."
l142: Please rate your NHE assessment according to the ANHEQ criteria (The ANHEQ evaluation criteria - Introducing reliable rating scales for Assessing Nordic Hamstring Exercise Quality) and add information about missing aspects such as feedback of movement speed or specific criteria for exclusion or repetition of trials
-> afterwards you can check the quality of your methodology with the one of published NHE assessments (Quo vadis Nordic Hamstring Exercise-related research - A scoping review revealing the need for improved methodology and reporting)
l155: this is not the original reference. This must be Hopkins or someone else ...
l158-160: This sentence is not correct ... Indeed, the SEM and MDC quantified the variance of participants' performance. Please consider adding calibration data of the force cells to check if the gathered data was valid.
l161: [22]:
-> Did any of the participants use additional loading?
RESULTS:
- please present a graph with representative force-time histories of different participants (e.g. S11 vs. S16 or S3 vs. S4) The first pair is interesting because S11 is much stronger, but S16 has a higher RFD. The latter pair has similar peak force, but different RFD. -> You must include a clearer picture of teh distinction between these parameters. Once again, right now RFD seems to be useless.
Fig2: please indicate that these are bilateral peak forces -> here and throughout the entire manuscript
Fig2: Why did you not normalize to body mass?
Fig2: Is there a correlation between RFD and average movement speed?
Tab2: add "bilateral"
l193-196: three times "showed" ...
l198: "a.u." (see above)
DISCUSSION:
l214: "bilateral" eccentric ...
l216-221: four times "showed" ...
l224-234: this paragraph belongs into the introduction section as well as l236-238.
l242: "large variability in rapid muscle activation" stands in big contrast to the intended task "slow controlled NHE" -> this controversy must be removed or explained
l247: add "knee position" to the limitations (refer to ANHEQ criteria), potential effect of ankle hook tilt?
l254: validation in terms of what? -> please specify and expand!
perspectives: knee and hip kinematics? other exercises feasible such as NH plank (Modulating the Nordic Hamstring Exercsie fom 'zero to hero' - A stepwise progression explored in a high-performance athlete)
There are some paragraphs with lots of duplicates (e.g. showed).
Author Response
Dear reviewer, thank you very much for the thorough revision of the paper, and we believe that your comments and suggestions improved the quality of the paper.
Please find attached all the answers point by point.
We hope that the manuscript could be now suitable for publication.
Thank you.

Reviewer 2 Report
As it is stated in the Abstract, the paper is aimed “ to assess the reliability of a new dynamometer that measures maximal strength (MS) and rate of force development (RFD) during the NHE” while in the title the "Reliability of Maximal Strength and Peak Rate of Force during NHE" is mentioned. Therefore, the title and aim of the paper are contradictory. Since the “new dynamometer” mentioned in the Abstract is really the portable dynamometer (PD) Neuroexcelence® (Porto, Portugal) that is sold in many online and offline shops, it must be pointed out everywhere starting from the Abstract.
11) If the aim of the study is verification of the reliability of the PD Neuroexcelence (as it is stated in the Abstract), it is not a matter for research paper, otherwise the measurement data must be compared to the data measured on the same group of volunteers by the isokinetic test dynamometry that is the gold standard.
22) If the aim of the study is verification of the reliability of the MS and PRF for medical diagnostics etc. (as it is stated in the title), the results must be compared to other potential diagnostic parameters
The manuscript in the present structure has no novelty. The accuracy and reliability of the PD Neuroexcelence have been checked by the developers and manufacturers. In any case, double measurements of two parameters on a small group of a narrow-age group of volunteers prove the reliability of neither PD nor MS+PRF because the control measurements by other type dynamometers have not been carried out by the authors.
The introduction with a literature review on hamstring injuries must be replaced by the dynamometry measurements or other technique connected with SENSORS, because the other chapters of the manuscript have no any relation to diagnosis/treatment/rehabilitation of hamstring injuries. Therefore, the list of REFs must be reworked out.
Though other dynamometry tests are criticized in the Introduction as ones with “high cost“ while according to the online prices, the PD Neuroexcelenceis even more expensive, the aim of the study is not reached.
The Acknowledgments statement is unclear.
The paper has no scientific novelty.
Some minor corrections must be introduced.
Author Response

(The authors gave the same response as above.)

Round 2
Reviewer 1 Report
Dear authors,
thank you very much for your major changes. The current version is much better, but the RFD issue is still not addressed in a sufficient way. Please find my specific comments below.
GENERAL:
- Where is it shown that RFD is a better predictor of athletic performance? Please be more specific and add references!
- I do not agree that RFD should be "used cautiously as it is a controlled and slow movement". Emphasize and indicate where the RFDmax occurred in Fig. 4 & 5 (please mark the corresponding 50ms intervals). This illustration will improve the understanding and interpretation of this parameter. RFDmax is high when movement velocity changes and/or fluctuates. I have still big problems to relate this parameter to a "strength capacity". Right now, I fear that your RFDmax values are located elsewhere in the ROM.
- "measuring RFD in smaller muscle groups" --> the hamstrings are not small ...
- Fig. 5: Why are there such large differences in force (from 80 N to 220 N) at movement onset? Please mention in the methods how movement onset was determined.
- Where are the force sensors located? How much ankle hook tilt usually occurs (in degrees)?
- Please add to the methods that no additional loading was used.
- Please add an information how far your participants controlled the movement. Did their reach 45° knee flexion? Or even 30°? When did they loose the control (DWA angle)?
SPECIFIC:
- l44: [6] has nothing to do with running/sprinting. Please remove the citation.
- l49: [6] has nothing to do with performance. Please remove the citation. Instead, add a reference to "Nordic Hamstring Exercise training induces improved lower-limb swing phase mechanics and sustained strength preservation in sprinters" because this study is the only one that links NHE-induced increased sprint performance with superior late swing phase mechanics.
- l80: "provides reliable data of" instead of "is reliable to"
- l161-162: your method does not deserve 2 points for knee position because the knees are not placed on an edge/protrude --> Therefore, your study has a total of 8 points which is a "good" quality and much better than the majority of previously published studies (Quo vadis Nordic Hamstring Exercise-related research - A scoping review revealing the need for improved methodology and reporting). Please change accordingly.
- l182-184: Why did you not address my previous comments regarding these 2 sentences?
- Fig.2 (and entire manuscript): Normalization refers to force/body mass. Please explain why you indicated absolute force values.
- Fig.5: Why "Total (N)" in legend? And the title does not read well. Please modify ... what does "relation strength" mean?
- l289: A "rapid muscle activation capacity" is not needed when a slow controlled NHE is conducted. Please revise!
- [6] has nothing to do with these NHE variations. Instead add a reference to "Modulating the Nordic Hamstring Exercsie fom 'zero to hero' - A stepwise progression explored in a high-performance athlete". --> This study investigated NHE variations where a rapid muscle acitivation is needed, i.e. reactively-bouncing and decelerating exercises which elicited much higher peak moments than the standard NHE.
Author Response
REVIEWER #1
Dear reviewer,
thank you very much for your major changes. The current version is much better, but the RFD issue is still not addressed in a sufficient way. Please find my specific comments below.
Authors´ response: Authors would like to thank the reviewer for the suggestions and the thorough revisions. We really appreciate the time considered, and the detailed comments and suggestions provided by the reviewer. For this second round of revision, we used the blue color. Thank you very much.
GENERAL:
- Where is it shown that RFD is a better predictor of athletic performance? Please be more specific and add references!
Author´s response: The RFD is a measure of how fast an athlete can develop force. Therefore, improving an athlete’s RFD may make them more powerful as the individual can develop higher forces in a shorter period of time; also, developing a more powerful athlete may improve sports performance. Please find below some references (included in the manuscript) that highlight the importance of RFD in different sports:
-Aagaard P., Simonsen E.B., Andersen J.L., Magnusson P., and Dyhre-Poulsen P. (2002) Increased rate of force development and neural drive of human skeleton muscle following resistance training. J Appl Physiol 93: 1318-1326. [PubMed]
- Andersen, L. L., Andersen, J. L., Zebis, M. K., & Aagaard, P. (2010). Early and late rate of force development: differential adaptive responses to resistance training?. Scandinavian journal of medicine & science in sports, 20(1), e162-e169.[PubMed]
-Blazevich, A. J., Cannavan, D., Horne, S., Coleman, D. R., & Aagaard, P. (2009). Changes in muscle force-length properties affect the early rise of force in vivo. Muscle & nerve, 39(4), 512.[PubMed]
-Laffaye, G., & Wagner, P. (2013). Eccentric rate of force development determines jumping performance. Computer Methods in Biomechanics and Biomedical Engineering, 16(1), pp.82–83. [Link]
-Haff, GG, Carlock, JM, Hartman, MJ, Kilgore, JL, Kawamori, N, Jackson, JR, Morris, RT, Sands, WA, and Stone, MH. Force-time curve characteristics of dynamic and isometric muscle actions of elite women Olympic weightlifters. J Strength Cond Res 19: 741–748, 2005. [PubMed]
-Haff, GG, Stone, MH, O’Bryant, HS, Harman, E, Dinan, CN, Johnson, R, and Han, KH. Force-time dependent characteristics of dynamic and isometric muscle actions. J Strength Cond Res 11: 269– 272, 1997. [Link]
-Kawamori, N, Rossi, SJ, Justice, BD, Haff, EE, Pistilli, EE, O’Bryant, HS, Stone, MH, and Haff, GG. Peak force and rate of force development during isometric and dynamic mid-thigh clean pulls performed at various intensities. J Strength Cond Res 20: 483–491, 2006. [PubMed]
-Slawinski, J, Bonnefoy, A, Leveˆque, JM, Ontanon, G, Riquet, A, Dumas, R, and Che` ze, L. Kinematic and kinetic comparisons of elite and well-trained sprinters during sprint start. J Strength Cond Res 24(4): 896–905, 2010 [PubMed]
-Stone, MH, Sands, WA, Carlock, J, Callan, S, Dickie, D, Daigle, K, Cotton, J, Smith, SL, and Hartman, M. The importance of isometric maximum strength and peak rate-of-force development in sprint cycling. J Strength Cond Res 18(4): 878–884, 2004. [PubMed]
-Leary, BK, Statler, J, Hopkins, B, Fitzwater, R, Kesling, T, Lyon, J, Phillips, B, Bryner, RW, Cormie, P, and Haff, GG. The relationship between isometric force-time curve characteristics and club head speed in recreational golfers. J Strength Cond Res 26: 2685–2697, 2012. [PubMed]
- I do not agree that RFD should be "used cautiously as it is a controlled and slow movement". Emphasize and indicate where the RFDmax occurred in Fig. 4 & 5 (please mark the corresponding 50ms intervals). This illustration will improve the understanding and interpretation of this parameter. RFDmax is high when movement velocity changes and/or fluctuates. I have still big problems to relate this parameter to a "strength capacity". Right now, I fear that your RFDmax values are located elsewhere in the ROM.
Author´s response: Thank you for your comment. We agree that we should emphasize more the limitations of the apparatus to measure RFD. Therefore, we reworded the sentence and now it reads:
L 310-315: “Considering the moderate within-subjects correlation between test-retest, it is important to highlight that RFD assessments might be challenged and may need more time of familiarization with the test [31]; also, it is important to highlight that the NHE is not performed in an maximal isometric contraction, as it is traditional RFD evaluations. Therefore, due to controlled and slow movement in the NHE, the highest RFD may not occur at the beginning of exercise”.
Furthermore, we reworded the final part of the conclusions:
L 342-345 “This study shows that MS can be measured during the NHE using a portable dynamometer, but this should be performed with caution and previous familiarization due to the slow and controlled movement of the NHE that may not favor the attainment of the highest RFD at the beginning of exercise”.
We also added Figure 5 which represents the RFD along the repetition for clarity.
- “measuring RFD in smaller muscle groups” the hamstrings are not small ...
Author´s response: We agree. This has been reworded.
- Fig. 5: Why are there such large differences in force (from 80 N to 220 N) at movement onset? Please mention in the methods how movement onset was determined.
Author´s response: The movement onset was determined by a signal given by the software; however, we acknowledge that it is possible that some participants were already producing force when the dynamometer started the measurement. The data are presented from 0. It is possible that subject 5 was already producing some force when the dynamometer started. The movement onset was determined by counting down from 3 to 1 (information given by the software) and then the subject started performing the Nordic hamstring exercise.
The information was added to the text.
L 158-160 “The movement onset was determined by counting down from three to one (information given by the software), and then the participant started performing the NHE.”
- Where are the force sensors located? How much ankle hook tilt usually occurs (in degrees)?
Author´s response: Thank you to point this out. When starting the movement, the reading of the cell is correct, but in the middle of the exercise, the hook has a rotational movement of about 5 to 8 degrees, which is intentional, which is the adaptation of the hook to the athlete exercise, which can vary the angle from athlete to athlete. The manufacturers considered this read error to be negligible. A load of 100 kg with a hook rotated by 8 degrees corresponds to an error of +/- 1 kg. The Cell reading is 100g in 100g. This information was added to the methods.
L 124-129 “When starting the movement, the reading of the cell is correct, but in the middle of the exercise, the hook has a rotational movement of about 5 to 8 degrees, which is intentional, which is the adaptation of the hook to the athlete exercise, which can vary the angle from athlete to athlete. The manufacturers considered this read error to be negligible. A load of 100 kg with a hook rotated by 8 degrees corresponds to an error of +/- 1 kg. The Cell reading is 100g in 100g.”
- Please add to the methods that no additional loading was used.
Author´s response: This has been added to the methods.
L 167-168 “Participants controlled the movement until to lose control and stop dealing with it. No additional loading was used.”
- Please add an information how far your participants controlled the movement. Did their reach 45° knee flexion? Or even 30°? When did they lose the control (DWA angle)?
Author´s response: Thank you for the opportunity to clarify this. The participants controlled the movement until to lose the control of their body and fail to deal with it. We have amended to the manuscript.
L 167-168 “Participants controlled the movement until to lose the control and stop dealing with it. No additional loading was used.”
SPECIFIC:
- l44: [6] has nothing to do with running/sprinting. Please remove the citation.
Author´s response: This has been reworded.
- l49: [6] has nothing to do with performance. Please remove the citation. Instead, add a reference to “Nordic Hamstring Exercise training induces improved lower-limb swing phase mechanics and sustained strength preservation in sprinters” because this study is the only one that links NHE-induced increased sprint performance with superior late swing phase mechanics.
Author´s response: The reference was changed.
- l80: “provides reliable data of” instead of “is reliable to”
Authors´ response: Change has been made.
- l161-162: your method does not deserve 2 points for knee position because the knees are not placed on an edge/protrude Therefore, your study has a total of 8 points which is“a "g”od" quality and much better than the majority of previously published studies (Quo vadis Nordic Hamstring Exercise-related resear–h - A scoping review revealing the need for improved methodology and reporting). Please change accordingly.
Author´s response: Thanks for noting this. The change has been made.
L169-170: “According to the ANHEQ criteria [21], the total score for NHE quality was 8 points which is considered “good”.
- l182-184: Why did you not address my previous comments regarding these 2 sentences?
Authors´ response: We understand the reviewer’s point, but we believe that it is relevant to present the corresponding data.
- Fig.2 (and entire manuscript): Normalization refers to force/body mass. Please explain why you indicated absolute force values.
Author´s response: We apologize, but previously we did not understand the comment. Thanks for mentioning it again. This has now been added in the table of results.
- Fig.5: Why "Total (N)" in legend? And the title does not read well. Please modify ... what does "relation strength" mean?
Author´s response: We agree with the comment. The changes have been made.
- l289: A "rapid muscle activation capacity" is not needed when a slow controlled NHE is conducted. Please revise!
Author´s response: We agree with the reviewer´s opinion. The sentence has been deleted. We wanted to present "rapid muscle activation capacity" from the general concept of RFD and not NHE, but we agree it is corresponding in this context. Thanks for the opportunity to clarify this.
- [6] has nothing to do with these NHE variations. Instead add a reference to "Modulating the Nordic Hamstring Exercsie fom 'zero to hero' - A stepwise progression explored in a high-performance athlete". --> This study investigated NHE variations where a rapid muscle acitivation is needed, i.e. reactively-bouncing and decelerating exercises which elicited much higher peak moments than the standard NHE
Author´s response: Thank you for pint this. The reference was added.
We would like to thank you for the thorough revision of the paper, and we believe that the reviewer's comments and suggestions improved significantly the quality of the paper. We hope that the manuscript could be now suitable for publication. Thank you very much.

Reviewer 2 Report
The authors corrected the manuscript according to my remarks, and now it can be accepted.
Author Response
Dear reviewer,
thank you, once again, for the opportunity to submit our work to a such recognized journal.
We would like to thank you for the thorough revision of the paper and for the acceptance of the current version of the work.
Thank you very much.
Round 3
Reviewer 1 Report
Dear authors,
once again I would congratulate for your major changes. There are some minor points before I can accept your manuscript for publication.
- For sure, RFDmax might be essential for sports performance. However, not all exercises are suitable to apply and determine this parameter! Please be cautious to use a "one fits it all-solution".
- Abstract: please add a sentence to the conclusion that the use of RFDmax in NHE is not straightforward (or can you answer the following questions: Where should RFDmax occur? How can it be enhanced?), although being some sort of reliable ... In my opinion, just being reliable does not mean that a parameter has an significant informative value for coaches and athletes.
- l190-192: These 2 sentences are not correct. These measures do not assess the measurement error and sensibility of the dynamometer, but the variability of the participants' performance!
- Fig. 4: Thanks for this graph which demonstrates that there is no clear pattern (in contrast to an isometric assessment of RFD). I still doubt the value and meaningfulness of RFDmax during slow controlled NHE!
- l314-315: It occurs at an arbitrary joint angle where there is a fluctuation of force application. There is no pattern where and why RFDmax is generated during a slow controlled NHE. Please expand how RFDmax can be enhanced or why! In my opinion, RFDmax during NHE simply depends on movement velocity and body mass. Consequently, there you should have normalized all your data on body mass ...
Sorry for the next round, but in the end I will prevent that RFD is simply used because it can be applied. Please take position to RFDmax dependence on body mass and movement velocity.
The reviewer
Author Response
REVIEWER #1
Dear authors,
once again I would congratulate for your major changes. There are some minor points before I can accept your manuscript for publication.
Authors´ response: Authors would like to thank the reviewer for the suggestions and the thorough revisions. We really appreciate the time considered, and the detailed comments and suggestions provided by the reviewer in all revisions. For this third round of revision, we used the green color. Thank you very much for the great learning.
- For sure, RFDmax might be essential for sports performance. However, not all exercises are suitable to apply and determine this parameter! Please be cautious to use a "one fits it all-solution".
Authors´ response: We totally agree with the reviewer. We added the following sentence to the manuscript. L 30-31: “However, not all exercises are suitable to apply and determine RFD, so caution must be given when analyzing RFD during NHE.”
- Abstract: please add a sentence to the conclusion that the use of RFDmax in NHE is not straightforward (or can you answer the following questions: Where should RFDmax occur? How can it be enhanced?), although being some sort of reliable ... In my opinion, just being reliable does not mean that a parameter has an significant informative value for coaches and athletes.
Authors´ response: We totally agree with the reviewer. Please see the previous answer. Thank you.
- l190-192: These 2 sentences are not correct. These measures do not assess the measurement error and sensibility of the dynamometer, but the variability of the participants' performance!
Authors´ response: We totally agree with the reviewer. It was changed accordingly.
L 191-192: “The standard error of measurement [25] and the minimal detectable change (MDC) were calculated to analyse the variability of the participants' performance.”
- Fig. 4: Thanks for this graph which demonstrates that there is no clear pattern (in contrast to an isometric assessment of RFD). I still doubt the value and meaningfulness of RFDmax during slow controlled NHE!
Authors´ response: We agree with the reviewer´s opinion, that there is no clear pattern for this metric, and for this reason we wanted to present the graph (be well-presented) as well as continuously stated throughout the manuscript (clearly documented) that RFD should be used with caution. Thank you.
- l314-315: It occurs at an arbitrary joint angle where there is a fluctuation of force application. There is no pattern where and why RFDmax is generated during a slow controlled NHE. Please expand how RFDmax can be enhanced or why! In my opinion, RFDmax during NHE simply depends on movement velocity and body mass. Consequently, there you should have normalized all your data on body mass ...
Authors´ response: Once again, we agree with the reviewer´s opinion and for this reason we already stated in the limitation, line 322-325 about “… the lack the determination of “optimal” angle peak torque of the knee flexor muscle group, which would be useful when targeting strength improvements at a specific joint angle (that could be measured by the gold standard measurement such as an isokinetic dynamometer)”. Regarding normalized all the data we think it is good very good idea for the following research about the effects of movement velocity and body mass in NHE test and metrics.
Sorry for the next round, but in the end I will prevent that RFD is simply used because it can be applied. Please take position to RFDmax dependence on body mass and movement velocity.
The reviewer
Author´s response: We would like to thank you for all revisions that was made for the paper, and we truly believe that the reviewer's comments and suggestions were/are very important and significantly improved the quality of the paper. Thank you very much for all the help and great learning during all this process. Best regards.
